# GMSF: Global Matching Scene Flow

**Yushan Zhang**   **Johan Edstedt**   **Bastian Wandt**
**Per-Erik Forssén**   **Maria Magnusson**   **Michael Felsberg**
Linköping University
{firstname.lastname}@liu.se

## Abstract

We tackle the task of scene flow estimation from point clouds. Given a source and a target point cloud, the objective is to estimate a translation from each point in the source point cloud to the target, resulting in a 3D motion vector field. Previous dominant scene flow estimation methods require complicated coarse-to-fine or recurrent architectures as a multi-stage refinement. In contrast, we propose a significantly simpler single-scale one-shot global matching to address the problem. Our key finding is that reliable feature similarity between point pairs is essential and sufficient to estimate accurate scene flow. We thus propose to decompose the feature extraction step via a hybrid local-global-cross transformer architecture which is crucial to accurate and robust feature representations. Extensive experiments show that the proposed Global Matching Scene Flow (GMSF) sets a new state-of-the-art on multiple scene flow estimation benchmarks. On FlyingThings3D, with the presence of occlusion points, GMSF reduces the outlier percentage from the previous best performance of 27.4% to 5.6%. On KITTI Scene Flow, without any fine-tuning, our proposed method shows state-of-the-art performance. On the Waymo-Open dataset, the proposed method outperforms previous methods by a large margin. The code is available at https://github.com/ZhangYushan3/GMSF.

## 1 Introduction

Scene flow estimation is a popular computer vision problem with many applications in autonomous driving [31] and robotics [39]. With the development of optical flow estimation and the emergence of numerous end-to-end trainable models in recent years, scene flow estimation, as a close research area to optical flow estimation, takes advantage of the rapid growth. As a result, many end-to-end trainable models have been developed for scene flow estimation using optical flow architectures [27, 46, 55]. Moreover, with the growing popularity of Light Detection and Ranging (LiDAR), the interest has shifted to computing scene flow from point clouds instead of stereo image sequences. In this work, we focus on estimating scene flow from 3D point clouds.

One of the challenges faced in scene flow estimation is fast movement. Previous methods usually employ a complicated multi-stage refinement with either a coarse-to-fine architecture [55] or a recurrent architecture [46] to address the problem. We instead propose to solve scene flow estimation by a single-scale one-shot global matching method, that is able to capture arbitrary correspondence, thus, handling fast movements. Occlusion is yet another challenge faced in scene flow estimation. We take inspiration from an optical flow estimation method [56] to enforce smoothness consistency during the matching process.

The proposed method consists of two stages: feature extraction and matching. A detailed description is given in Section 3. To extract high-quality features, we take inspiration from the recently dominant transformers [47] and propose a hybrid local-global-cross transformer architecture to learn accurate and robust feature representations. Both local and global-cross transformers are crucial for our approach as also shown experimentally in Section 4.5. The global matching process, including

37th Conference on Neural Information Processing Systems (NeurIPS 2023).

estimation and refinement, is guided solely by feature similarity matrices. First, scene flow is calculated as a weighted average of translation vectors from each source point to all target points under the guidance of a cross-feature similarity matrix. Since the matching is done in a global manner, it can capture short-distance correspondences as well as long-distance correspondences and, therefore, is capable of dealing with fast movements. Further refinement is done under the guidance of a self-feature similarity matrix to ensure scene flow smoothness in areas with similar features. This allows to propagate the estimated scene flow from non-occluded areas to occluded areas, thus solving the problem of occlusions.

To summarize, our contributions are: (1) A hybrid local-global-cross transformer architecture is introduced to learn accurate and robust feature representations of 3D point clouds. (2) Based on the similarity of the hybrid features, we propose to use a global matching process to solve the scene flow estimation. (3) Extensive experiments on popular datasets show that the proposed method outperforms previous scene flow methods by a large margin on FlyingThings3D [30] and Waymo-Open [44] and achieves state-of-the-art generalization ability on KITTI Scene Flow [31].

## 2 Related Work

### 2.1 Scene Flow

Scene flow estimation [23] has developed quickly since the introduction of the KITTI Scene Flow [31] and FlyingThings3D [30] benchmarks, which were the first benchmarks for estimating scene flow from stereo videos. Many scene flow methods [1, 29, 31, 37, 40, 48, 58] assume that the objects in a scene are rigid and decompose the estimation task into subtasks. These subtasks often involve first detecting or segmenting objects in the scene and then fitting motion models for each object. In autonomous driving scenes, these methods are often effective, as such scenes typically involve static backgrounds and moving vehicles. However, they are not capable of handling more general scenes that include deformable objects. Moreover, the subtasks introduce non-differentiable components, making end-to-end training impossible without instance-level supervision.

Recent work in scene flow estimation mostly takes inspiration from the related task of optical flow [9, 16, 41, 45] and can be divided into several categories: encoder-decoder methods [14, 27] that solve the scene flow by an hourglass architecture neural network, multi-scale methods [3, 20, 55] that estimate the motion from coarse to fine scales, or recurrent methods [17, 46, 53] that iteratively refine the estimated motion. Other approaches [19, 34] try to solve the problem by finding soft correspondences on point pairs within a small region. In order to reduce the annotation requirement, some methods focus on runtime optimization [22, 18], prior assumptions [21], or even without the need for training data [4].

**Encoder-decoder Methods:** Flownet [9] and Flownet2.0 [16], were the first methods to learn optical flow end-to-end with an hourglass-like model, and inspired many later methods. Flownet3D [27] first employs a set of convolutional layers to extract coarse features. A flow embedding layer is introduced to associate points based on their spatial localities and geometrical similarities on a coarse scale. A set of upscaling convolutional layers is then introduced to upsample the flow to the high resolution. FlowNet3D++ [52] further incorporates point-to-plane distance and angular distance as additional geometry constraints to Flownet3D [27]. HPLFlowNet [14] employs Bilateral Convolutional Layers (BCL) to restore structural information from unstructured point clouds. Following the hourglass-like model, DownBCL, UpBCL, and CorrBCL operations are proposed to restore information from each point cloud and fuse information from both point clouds.

**Coarse-to-fine Methods:** PointPWC-Net [55] is a coarse-to-fine method for scene flow estimation using hierarchical feature extraction and warping, which is based on the optical flow method PWC-Net [41]. A novel learnable Cost Volume Layer is introduced to aggregate costs in a patch-to-patch manner. Additional self-supervised losses are introduced to train the model without ground-truth labels. Bi-PointFlowNet [3] follows the coarse-to-fine scheme and introduces bidirectional flow embedding layers to learn features along both forward and backward directions. Based on previous methods [27, 55], HCRF-Flow [20] introduces a high-order conditional random fields (CRFs) based relation module (Con-HCRFs) to explore rigid motion constraints among neighboring points to force point-wise smoothness and within local regions to force region-wise rigidity. FH-Net [7] proposes a

fast hierarchical network with lightweight Trans-flow layers to compute key points flow and inverse Trans-up layers to upsample the coarse flow based on the similarity between sparse and dense points.

**Recurrent Methods:**    FlowStep3D [17], is the first recurrent method for non-rigid scene flow estimation. They first use a global correlation unit to estimate an initial flow at the coarse scale, and then update the flow iteratively by a Gated Recurrent Unit (GRU). RAFT3D [46] also adopts a recurrent framework. Here, the objective is not the scene flow itself but a dense transformation field that maps each point from the first frame to the second frame. The transformation is then iteratively updated by a GRU. PV-RAFT [53] presents point-voxel correlation fields to capture both short-range and long-range movements. Both coarse-to-fine and recurrent methods take the cost volume as input to a convolutional neural network for scene flow prediction. However, these regression techniques may not be able to accurately capture fast movements, and as a result, multi-stage refinement is often necessary. On the other hand, we propose a simpler architecture that solves scene flow estimation in a single-scale global matching process with no iterative refinement.

**Soft Correspondence Methods:**    Some work poses the scene flow estimation as an optimal transport problem. FLOT [34] introduces an Optimal Transport Module that gives a dense transport plan informing the correspondence between all pairs of points in the two point clouds. Convolutional layers are further applied to refine the scene flow. SCTN [19] introduces a voxel-based sparse convolution followed by a point transformer feature extraction module. Both features, from convolution and transformer, are used for correspondence computation. However, these methods involve complicated regularization and constraints to estimate the optimal transport from the correlation matrix. Moreover, the correspondences are only computed within a small neighboring region. We instead follow the recent global matching paradigm [10, 56, 64] and solve the scene flow estimation with a global matcher that is able to capture both short-distance and long-distance correspondence.

**Runtime Optimization, Prior Assumptions, and Self-supervision:**    Different from the proposed method, which is fully supervised and trained offline, some other work focuses on runtime optimization, prior assumptions, and self-supervision. Li *et al.* [22] revisit the need for explicit regularization in supervised scene flow learning. The deep learning methods tend to rely on prior statistics learned during training, which are domain-specific. This does not guarantee generalization ability during testing. To this end, Li *et al.* propose to rely on runtime optimization with scene flow prior as strong regularization. Based on [22] Lang *et al.* [18] propose to combine runtime optimization with self-supervision. A correspondence model is first trained to initialize the flow. Refinement is done by optimizing the flow refinement component during runtime. The whole process can be done under self-supervision. Pontes *et al.* [33] propose to use the graph Laplacian of a point cloud to force the scene flow to be "as rigid as possible". Same as in [22], this constraint can be optimized during runtime. Li *et al.* [21] propose a self-supervised scene flow learning approach with local rigidity prior assumption for real-world scenes. Instead of relying on point-wise similarities for scene flow estimation, region-wise rigid alignment is enforced. Most recently, Chodosh *et al.* [4] identify the main challenges of LiDAR scene flow estimation as estimating the remaining simple motions after removing the dominant rigid motion. By combining ICP, rigid assumptions, and runtime optimization, they achieve state-of-the-art performance without any training data.

## 2.2   Point Cloud Registration

Related to scene flow estimation, there are some correspondence-based point cloud registration methods. Such methods separate the point cloud registration task into two stages: finding the correspondences and recovering the transformation. PPFNet [6] and PPF-FoldNet [5] proposed by Deng *et al.* focus on finding sparse corresponding 3D local features. Gojcic *et al.* [12] propose to use voxelized smoothed density value (SDV) representation to match 3D point clouds. These methods only compute sparse correspondences and are not capable of handling dense correspondences required in scene flow tasks. More related works are CoFiNet [59] and GeoTransformer [36], both of which involve finding dense correspondences employing transformer architectures. Yu *et al.* in CoFiNet [59] propose a detection-free learning framework and find dense point correspondence in a coarse-to-fine manner. Qin *et al.* in GeoTransformer [36] further improve the accuracy by leveraging the geometric information. RoITr [60] introduces a Rotation-Invariant Transformer to disentangle the geometry and poses, and tackle point cloud matching under arbitrary pose variations. PEAL [61] introduces the Prior Embedded Explicit Attention Learning model (PEAL), and for the first time explicitly injects

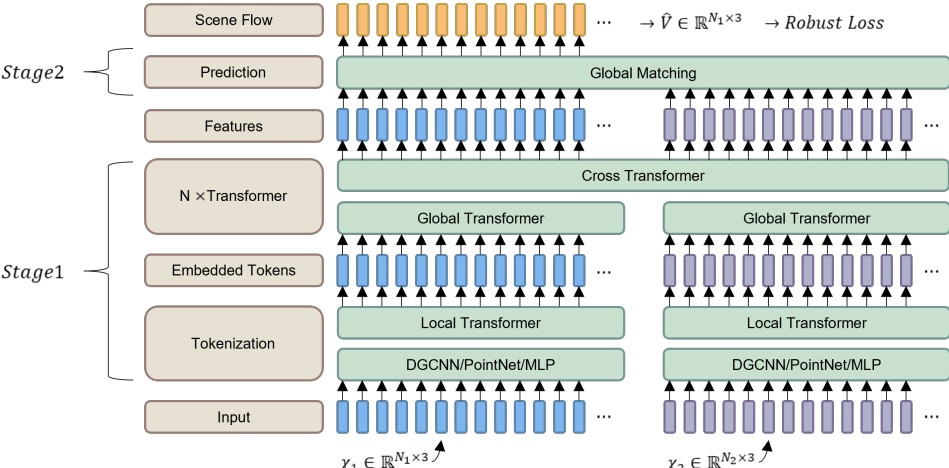

Figure 1: **Method Overview.** We propose a simple yet powerful method for scene flow estimation. In the first stage (see Section 3.1) we propose a strong local-global-cross transformer architecture that is capable of extracting robust and highly localizable features. In the second stage (see Section 3.2), a simple global matching yields the flow. In comparison to previous work, our approach is significantly simpler, while achieving state-of-the-art results.

overlap prior into Transformer to solve point cloud registration under low overlap. However, the goal of point cloud registration is not to estimate the translation vectors for each of the points, which makes our work different from these approaches.

## 2.3 Transformers

Transformers were first proposed in [47] for translation tasks with an encoder-decoder architecture using only attention and fully connected layers. Transformers have been proven to be efficient in sequence-to-sequence problems, well-suited to research problems involving sequential and unstructured data. The key to the success of transformers over convolutional neural networks is that they can capture long-range dependencies within the sequence, which is very important, not only in translation but also in many other tasks e.g. computer vision [8], audio processing [24], recommender systems [42], and natural language processing [54].

Transformers have also been explored for point clouds [28]. The coordinates of all points are stacked together directly as input to the transformers. For the tasks of classification and segmentation, PT [63] proposes constructing a local point transformer using k-nearest-neighbors. Each of the points would then attend to its nearest neighbors. PointASNL [57] uses adaptive sampling before the local transformer, and can better deal with noise and outliers. PCT [15] proposes to use global attention and results in a global point transformer. Pointformer [32] proposes a new scheme where first local transformers are used to extract multi-scale feature representations, then local-global transformers are used as cross attention to multi-scale features, finally, a global transformer captures context-aware representations. Point-BERT [62] is originally designed for masked point modeling. Instead of treating each point as one data item, they group the point cloud into several local patches. Each of these sub-clouds is tokenized to form input data.

Previous work on scene flow estimation exploits the capability of transformers for feature extraction either using global-based transformers in a local matching paradigm [19] or local-based transformers in a recurrent architecture [11]. Instead, we propose to leverage both local and global transformers to learn a feature representation for each point on a single scale. We show that high-quality feature representations are the fundamental property that is needed for scene flow estimation when formulated as a global matching problem.

# 3 Proposed Method

Given two point clouds $\mathcal{X}_1 \in \mathbb{R}^{N_1 \times 3}$ and $\mathcal{X}_2 \in \mathbb{R}^{N_2 \times 3}$ with only position information, the objective is to estimate the *scene flow* $V \in \mathbb{R}^{N_1 \times 3}$ that maps each point in the source point cloud to the target point cloud. Due to the sparse nature of the point clouds, the points in the source and the target point clouds do not necessarily have a one-to-one correspondence, which makes it difficult to formulate scene flow estimation as a dense matching problem. Instead, we show that learning a cross-feature similarity matrix of point pairs as soft correspondence is sufficient for scene flow estimation. Unlike many applications based on point cloud processing which need to acquire a high-level understanding, e.g. classification and segmentation, scene flow estimation requires a low-level understanding to distinguish geometry features between each element in the point clouds. To this end, we propose a transformer architecture to learn high-quality features for each point. The proposed method consists of two core components: feature extraction (see Section 3.1) and global matching (see Section 3.2). The overall framework is shown in Figure 1.

## 3.1 Feature Extraction

**Tokenization:** Given the 3D point clouds $\mathcal{X}_1$, $\mathcal{X}_2$, each point $x_i$ is first tokenized to get summarized information of its local neighbors. We first employ an off-the-shelf feature extraction network DGCNN [51] to map the input 3D coordinate $x_i$ into a high dimensional feature space $x_i^h$ conditioned on its nearest neighbors $x_j$. Each layer of the network can be written as

$$x_i^h = \max_{x_j \in \mathcal{N}(i)} h(x_i, x_j - x_i),$$
(1)

where $h$ represents a sequence of linear layers, batch normalization, and ReLU layers. The local neighbors $x_j \in \mathcal{N}(i)$ are found by a k-nearest-neighbor (knn) algorithm. Multiple layers are stacked together to get the final feature representation.

For each point, local information is incorporated within a small region by applying a local Point Transformer [63] within $x_j \in \mathcal{N}(i)$. The transformer is given by

$$x_i^l = \sum_{x_j \in \mathcal{N}(i)} \gamma(\varphi_l(x_i^h) - \psi_l(x_j^h) + \delta) \odot (\alpha_l(x_j^h) + \delta),$$
(2)

where the input features are first passed through linear layers $\varphi_l$, $\psi_l$, and $\alpha_l$ to generate query, key and value. $\delta$ is the relative position embedding that gives information about the 3D coordinate distance between $x_i$ and $x_j$. $\gamma$ represents a Multilayer Perceptron consisting of two linear layers and one ReLU nonlinearity. The output $x_i^l$ is further processed by a linear layer and a residual connection from $x_i^h$.

**Global-cross Transformer:** Transformer blocks are used to process the embedded tokens. Each of the blocks includes self-attention followed by cross-attention [38, 43, 47, 56].

The self-attention is formulated as

$$x_i^g = \sum_{x_j \in \mathcal{X}_1} \langle \varphi_g(x_i^l), \psi_g(x_j^l) \rangle \alpha_g(x_j^l),$$
(3)

where each point $x_i \in \mathcal{X}_1$ attends to all the other points $x_j \in \mathcal{X}_1$, same for the points $x_i \in \mathcal{X}_2$. Linear layers $\varphi_g$, $\psi_g$, and $\alpha_g$ generate the query, key, and value. $\langle , \rangle$ denotes a scalar product. Linear layer, layer norm, and skip connection are further applied to complete the self-attention module.

The cross-attention is given as

$$x_i^c = \sum_{x_j \in \mathcal{X}_2} \langle \varphi_c(x_i^g), \psi_c(x_j^g) \rangle \alpha_c(x_j^g),$$
(4)

where each point $x_i \in \mathcal{X}_1$ in the source point cloud attends to all the points $x_j \in \mathcal{X}_2$ in the target point cloud, and vice versa. A Feedforward network with multi-layer perceptron and layer norm is applied to aggregate information to the next transformer block. The detailed architecture of our proposed local-global-cross transformer is presented in Figure 2. The feature matrices $F_1 \in \mathbb{R}^{N_1 \times d}$ and $F_2 \in \mathbb{R}^{N_2 \times d}$ are formed as the concatenation of all the output feature vectors from the final transformer block, where $N_1$ and $N_2$ are the number of points in the two point clouds and $d$ is the feature dimension.

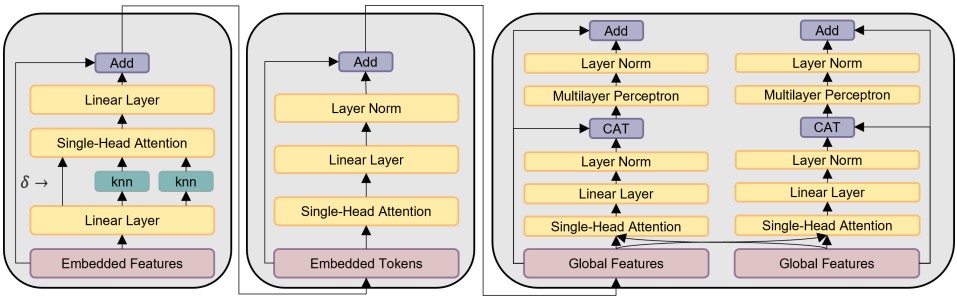

Figure 2: **Transformer Architecture**. Detailed local (left), global (middle), and cross (right) transformer architecture. The local transformer incorporates attention within a small number of neighbors. The global transformer is applied on the source and target points separately and incorporates attention on the whole point clouds. The cross transformer further attends to the other point cloud and gets the final representation conditioned on both the source and the target.

## 3.2 Global Matching

Feature similarity matrices are the only information that is needed for an accurate scene flow estimation. First, the *cross similarity matrix* between the source and the target point clouds is given by multiplying the feature matrices $F_1$ and $F_2$ and then normalizing over the second dimension with softmax to get a right stochastic matrix,

$$C_{\text{cross}} = \frac{F_1 F_2^T}{\sqrt{d}}, \tag{5}$$

$$M_{\text{cross}} = \text{softmax}(C_{\text{cross}}), \tag{6}$$

where each row of the matrix $M_{\text{cross}} \in \mathbb{R}^{N_1 \times N_2}$ is the matching confidence from one point in the source point cloud to all the points in the target point cloud. The second similarity matrix is the *self similarity matrix* of the source point cloud, given by

$$C_{\text{self}} = \frac{W_q(F_1) W_k(F_1)^T}{\sqrt{d}}, \tag{7}$$

$$M_{\text{self}} = \text{softmax}(C_{\text{self}}), \tag{8}$$

which is a matrix multiplication of the linearly projected point feature $F_1$. $W_q$ and $W_k$ are learnable linear projection layers. Each row of the matrix $M_{\text{self}} \in \mathbb{R}^{N_1 \times N_1}$ is the feature similarity between one point in the source point cloud to all the other points in the source point cloud. Given the point cloud coordinates $\mathcal{X}_1 \in \mathbb{R}^{N \times 3}$ and $\mathcal{X}_2 \in \mathbb{R}^{N \times 3}$, the estimated matching point $\hat{\mathcal{X}}_2$ in the target point cloud is computed as a weighted average of the 3D coordinates based on the matching confidence

$$\hat{\mathcal{X}}_2 = M_{\text{cross}} \mathcal{X}_2. \tag{9}$$

The scene flow is computed as the movement between the matching points

$$\hat{V}_{\text{inter}} = \hat{\mathcal{X}}_2 - \mathcal{X}_1. \tag{10}$$

The estimation procedure can also be seen as a weighted average of the translation vectors between point pairs, where a softmax ensures that the weights sum to one.

For occlusions in the source point cloud, the matching would fail under the assumption that there exists a matching point in the target point cloud. We avoid this by employing a self similarity matrix that utilizes information from the source point cloud. The self similarity matrix $M_{\text{self}}$ bears the similarity information for each pair of points in the source point cloud. Nearby points tend to share similar features and thus have higher similarities. Multiplying $M_{\text{self}}$ with the predicted scene flow $\hat{V}_{\text{inter}}$ can be seen as a smoothing procedure, where for each point, its predicted scene flow vector is updated as the weighted average of the scene flow vectors of the nearby points that share similar features. This also allows the network to propagate the correctly computed non-occluded scene flow estimation to its nearby occluded areas, which gives

$$\hat{V}_{\text{final}} = M_{\text{self}} \hat{V}_{\text{inter}}. \tag{11}$$

## 3.3 Loss Formulation

Let $\hat{V}$ be the estimated scene flow and $V_{gt}$ be the ground truth. We follow CamLiFlow [25] and use a robust training loss to supervise the process, given by

$$\mathcal{L}_{\hat{V}} = \sum_i (\|\hat{V}_{\text{final}}(i) - V_{\text{gt}}(i)\|_1 + \epsilon)^q, \tag{12}$$

where $\epsilon$ is set to 0.01 and $q$ is set to 0.4.

## 4 Experiments

### 4.1 Implementation Details

The proposed method is implemented in PyTorch. Following previous methods [14, 55], the numbers of points $N_1$ and $N_2$ are both set to 8192 during training and testing, randomly sampled from the full set. We perform data augmentation by randomly flipping horizontally and vertically. We use the AdamW optimizer with a learning rate of $2 \cdot 10^{-4}$, a weight decay of $10^{-4}$, and OneCycleLR as the scheduler to anneal the learning rate. The training is done for 600k iterations with a batch size of 8.

### 4.2 Evaluation Metrics

For a fair comparison we follow previous work [14, 46, 55] and evaluate the proposed method with the accuracy metric $EPE_{3D}$, and the robustness metrics $ACC_S$, $ACC_R$ and $Outliers$. $EPE_{3D}$ is the 3D end point error $\| \hat{V} - V_{gt} \|_2$ between the estimated scene flow and the ground truth averaged over each point. $ACC_S$ is the percentage of the estimated scene flow with an end point error less than 0.05 meter or relative error less than 5%. $ACC_R$ is the percentage of the estimated scene flow with an end point error less than 0.1 meter or relative error less than 10%. $Outliers$ is the percentage of the estimated scene flow with an end point error more than 0.3 meter or relative error more than 10%.

### 4.3 Datasets

The proposed method is tested on three established benchmarks for scene flow estimation.

**FlyingThings3D** [30] is a synthetic dataset of objects generated by ShapeNet [2] with randomized movement rendered in a scene. The dataset consists of 25000 stereo frames with ground truth data.

**KITTI Scene Flow** [31] is a real world dataset for autonomous driving. The annotation is done with the help of CAD models. It consists of 200 scenes for training and 200 scenes for testing.

Both datasets have to be preprocessed in order to obtain 3D points from the depth images. There exist two widely used preprocessing methods to generate the point clouds and the ground truth scene flow, one proposed by Liu *et al.* in FlowNet3D [27] and the other proposed by Gu *et al.* in HPLFlowNet [14]. The difference between the two approaches is that Liu *et al.* [27] keeps all valid points with an occlusion mask available during training and testing. Gu *et al.* [14] simplifies the task by removing all occluded points. We denote the datasets preprocessed by Liu *et al.* in FlowNet3D as F3D$_o$/KITTI$_o$ and by Gu *et al.* in HPLFlowNet as F3D$_s$/KITTI$_s$. In the original setting from [14, 27], the FlyingThing3D dataset **F3D$_s$** consists of 19640 and 3824 stereo scenes for training and testing, respectively. **F3D$_o$** consists of 20000 and 2000 stereo scenes for training and testing, respectively. For the KITTI dataset, **KITTI$_s$** consists of 142 scenes from the training set, and **KITTI$_o$** consists of 150 scenes from the training set. Since there is no annotation available in the testing set of KITTI, we follow previous methods to test the generalization ability of the proposed method without any fine-tuning on KITTI$_s$ and KITTI$_o$. For better evaluation and analysis, we additionally follow the setting in CamLiFlow [25] to extend F3D$_s$ to include occluded points. We denote this as **F3D$_c$**.

**Waymo-Open Dataset** [44] is a large-scale autonomous driving dataset. We follow [7] to preprocess the dataset to create the scene flow dataset. The dataset contains 798 training and 202 validation sequences. Each sequence consists of 20 seconds of 10Hz point cloud data. Different from [7] which only contains 100 sequences, we trained and tested our model on the full dataset.

## 4.4 State-of-the-art Comparison

We compare our proposed method GMSF with state-of-the-art methods on FlyingThings3D in different settings. Table 1 shows the results on F3D$_c$. Evaluation metrics are calculated over both *non-occluded* points and *all* points. Among all the methods, including methods with the corresponding stereo images as additional input [46], or even with optical flow as additional ground truth for supervision [25, 26], our proposed method achieves the best performance both in terms of accuracy and robustness.

To give a fair comparison with previous methods we report results on F3D$_o$ and F3D$_s$ with generalization to KITTI$_o$ and KITTI$_s$ in Table 2 and Table 3. The proposed method achieves the best performance on both F3D$_o$ and F3D$_s$, surpassing other state-of-the-art methods by a large margin. The generalization ability of the proposed model on KITTI$_o$ and KITTI$_s$ also achieves state of the art.

We further conduct experiments on the Waymo-Open dataset. We train and test on the full dataset with 798 training and 202 testing sequences. Comparisons with state of the art are given in Table 4.

Table 1: **State-of-the-art comparison on F3D$_c$.** The input modalities are given as a reference. Our method with only 3D points as input outperforms all the other state-of-the-art methods on all metrics.

| Method | Input | $EPE_{3D} \downarrow$ non-occluded | $ACC_S \uparrow$ non-occluded | $EPE_{3D} \downarrow$ all | $ACC_S \uparrow$ all |
|---|---|---|---|---|---|
| FlowNet3D [27] CVPR'19 | Points | 0.158 | 22.9 | 0.214 | 18.2 |
| RAFT3D [46] CVPR'21 | Image+Depth | - | - | 0.094 | 80.6 |
| CamLiFlow [25] CVPR'22 | Image+Points | 0.032 | 92.6 | 0.061 | 85.6 |
| CamLiPWC [26] arxiv'23 | Image+Points | - | - | 0.057 | 86.3 |
| CamLiRAFT [26] arxiv'23 | Image+Points | - | - | 0.049 | 88.4 |
| **GMSF(ours)** | Points | **0.022** | **95.9** | **0.040** | **92.6** |

Table 2: **State-of-the-art comparison on F3D$_o$ and KITTI$_o$.** The models are only trained on F3D$_o$ prepared by [27] with occlusions. Testing results on F3D$_o$ and KITTI$_o$ are given.

| Method | F3D$_O$ | | | | KITTI$_O$ | | | |
|---|---|---|---|---|---|---|---|---|
| | $EPE_{3D} \downarrow$ | $ACC_S \uparrow$ | $ACC_R \uparrow$ | $Outliers \downarrow$ | $EPE_{3D} \downarrow$ | $ACC_S \uparrow$ | $ACC_R \uparrow$ | $Outliers \downarrow$ |
| FlowNet3D [27] | 0.157 | 22.8 | 58.2 | 80.4 | 0.183 | 9.8 | 39.4 | 79.9 |
| HPLFlowNet [14] | 0.168 | 26.2 | 57.4 | 81.2 | 0.343 | 10.3 | 38.6 | 81.4 |
| PointPWC [55] | 0.155 | 41.6 | 69.9 | 63.8 | 0.118 | 40.3 | 75.7 | 49.6 |
| FLOT [34] | 0.153 | 39.6 | 66.0 | 66.2 | 0.130 | 27.8 | 66.7 | 52.9 |
| CamLiPWC [26] | 0.092 | 71.5 | 87.1 | 37.2 | - | - | - | - |
| CamLiRAFT [26] | 0.076 | 79.4 | 90.4 | 27.9 | - | - | - | - |
| Bi-PointFlow [3] | 0.073 | 79.1 | 89.6 | 27.4 | 0.065 | 76.9 | 90.6 | 26.4 |
| RAFT3D [46] | 0.064 | 83.7 | 89.2 | - | - | - | - | - |
| 3DFlow [49] | 0.063 | 79.1 | 90.9 | 27.9 | 0.073 | 81.9 | 89.0 | 26.1 |
| SCOOP+ [18] | - | - | - | - | 0.047 | 91.3 | 95.0 | 18.6 |
| **GMSF(ours)** | **0.022** | **95.0** | **97.5** | **5.6** | **0.033** | **91.6** | **95.9** | **13.7** |

## 4.5 Ablation Study

Table 6 shows the results of different numbers of **global-cross transformer** layers. While our approach technically works even without global-cross transformer layers, the performance is significantly worse compared to using two or more layers. This shows that only incorporating local information for the feature representation is insufficient for global matching. Moreover, the capacity of the network improves with the number of layers and achieves the best performance at 10 layers.

Table 7 shows the importance of different components in the **tokenization** process. We tried different methods, DGCNN [51], PointNet [35], and MLP, to map the 3D coordinates of the points into the high-dimensional feature space. For each of these mapping methods, the influence of the Local Point Transformer [63] is tested. When the local transformer is present, the metrics are similar with different mapping strategies, which demonstrate the effectiveness of the proposed local-global-cross transformer architecture. In the absence of the local transformer, the performance remains comparable with DGCNN for mapping but drops significantly with PointNet or MLP, which indicates the necessity of local information encoded in the tokenization step.

Table 8 gives the ablation study on feature dimensions. The default number of feature dimensions is 128 in our model. Reducing the number of feature dimensions leads to a lack of capacity of the model.

Table 3: **State-of-the-art comparison on F3D$_s$ and KITTI$_s$.** The models are only trained on F3D$_s$ prepared by [14] without occlusions. Testing results on F3D$_s$ and KITTI$_s$ are given.

| Method | F3D$_S$ | | | | KITTI$_S$ | | | |
|---|---|---|---|---|---|---|---|---|
| | $EPE_{3D}\downarrow$ | $ACC_S\uparrow$ | $ACC_R\uparrow$ | $Outliers\downarrow$ | $EPE_{3D}\downarrow$ | $ACC_S\uparrow$ | $ACC_R\uparrow$ | $Outliers\downarrow$ |
| FlowNet3D [27] | 0.1136 | 41.25 | 77.06 | 60.16 | 0.1767 | 37.38 | 66.77 | 52.71 |
| HPLFlowNet [14] | 0.0804 | 61.44 | 85.55 | 42.87 | 0.1169 | 47.83 | 77.76 | 41.03 |
| PointPWC [55] | 0.0588 | 73.79 | 92.76 | 34.24 | 0.0694 | 72.81 | 88.84 | 26.48 |
| FLOT [34] | 0.0520 | 73.20 | 92.70 | 35.70 | 0.0560 | 75.50 | 90.80 | 24.20 |
| HCRF-Flow [20] | 0.0488 | 83.37 | 95.07 | 26.14 | 0.0531 | 86.31 | 94.44 | 17.97 |
| PV-RAFT [53] | 0.0461 | 81.69 | 95.74 | 29.24 | 0.0560 | 82.26 | 93.72 | 21.63 |
| FlowStep3D [17] | 0.0455 | 81.62 | 96.14 | 21.65 | 0.0546 | 80.51 | 92.54 | 14.92 |
| RCP [13] | 0.0403 | 85.67 | 96.35 | 19.76 | 0.0481 | 84.91 | 94.48 | 12.28 |
| SCTN [19] | 0.0380 | 84.70 | 96.80 | 26.80 | 0.0370 | 87.30 | 95.90 | 17.90 |
| CamLiPWC [26] | 0.0320 | 92.50 | 97.90 | 15.60 | - | - | - | - |
| CamLiRAFT [26] | 0.0290 | 93.00 | 98.00 | 13.60 | - | - | - | - |
| Bi-PointFlow [3] | 0.0280 | 91.80 | 97.80 | 14.30 | 0.0300 | 92.00 | 96.00 | 14.10 |
| 3DFlow [49] | 0.0281 | 92.90 | 98.17 | 14.58 | 0.0309 | 90.47 | 95.80 | 16.12 |
| PT-FlowNet [11] | 0.0304 | 91.42 | 98.14 | 17.35 | 0.0224 | 95.51 | **98.38** | 11.86 |
| **GMSF(ours)** | **0.0090** | **99.18** | **99.69** | **2.55** | **0.0215** | **96.22** | 98.25 | **9.84** |

Table 4: **State-of-the-art comparison on Waymo-Open dataset.**

| Method | $EPE_{3D}\downarrow$ | $ACC_S\uparrow$ | $ACC_R\uparrow$ | $Outliers\downarrow$ |
|---|---|---|---|---|
| FlowNet3D [27] | 0.225 | 23.0 | 48.6 | 77.9 |
| PointPWC [55] | 0.307 | 10.3 | 23.1 | 78.6 |
| FESTA [50] | 0.223 | 24.5 | 27.2 | 76.5 |
| FH-Net [7] | 0.175 | 35.8 | 67.4 | 60.3 |
| **GMSF(ours)** | **0.083** | **74.7** | **85.1** | **43.5** |

Table 5: **Meta-information.**

| | |
|---|---|
| Runtime(ms) | 417.3 |
| FLOPs(G) | 654.32 |
| Parameters(M) | 7.07 |
| Memory (test)(GB) | 4.99 |
| Memory (train)(GB) | 162.3 |

Table 6: **Ablation study on the number of global-cross transformer layers on F3D$_c$.** The influence of the number of global-cross transformer layers is tested. The best performance is gained at 10 transformer layers.

| Layers | $EPE_{3D}\downarrow$ all | $ACC_S\uparrow$ all | $ACC_R\uparrow$ all | $Outliers\downarrow$ all | $EPE_{3D}\downarrow$ non-occ | $ACC_S\uparrow$ non-occ | $ACC_R\uparrow$ non-occ | $Outliers\downarrow$ non-occ |
|---|---|---|---|---|---|---|---|---|
| 0 | 0.212 | 39.01 | 63.59 | 66.51 | 0.132 | 43.95 | 70.24 | 62.92 |
| 2 | 0.075 | 79.02 | 90.22 | 25.64 | 0.047 | 84.67 | 94.23 | 22.07 |
| 4 | 0.055 | 87.37 | 93.76 | 16.39 | 0.032 | 92.01 | 96.84 | 13.41 |
| 6 | 0.050 | 89.32 | 94.60 | 14.23 | 0.029 | 93.46 | 97.33 | 11.54 |
| 8 | 0.045 | 91.22 | 95.25 | 12.11 | 0.025 | 94.91 | 97.70 | 9.68 |
| 10 | **0.040** | **92.64** | **95.84** | **10.34** | **0.022** | **95.94** | **98.06** | **8.13** |
| 12 | 0.043 | 91.95 | 95.57 | 11.12 | 0.024 | 95.40 | 97.88 | 8.81 |
| 14 | 0.045 | 91.66 | 95.41 | 11.54 | 0.025 | 95.19 | 97.78 | 9.21 |
| 16 | 0.044 | 91.74 | 95.51 | 11.33 | 0.025 | 95.38 | 97.93 | 8.91 |

## 4.6 FLOPs, GPU memory, and Runtime.

We report meta-information on our experiments: runtime (ms per scene) during testing on an NVIDIA A40 GPU, FLOPs (G), Number of parameters (M), GPU memory (GB) during testing (batch size 1) and training (batch size 8) with 10 transformer layers and 128 feature dimensions in Table 5.

## 4.7 Visualization

Figure 3 shows a visualization of the GMSF results on two samples from the FlyingThings3D dataset. Red and blue points represent the source and the target point clouds, respectively. Green points

Table 7: **Ablation study on the components of tokenization on F3D$_c$.** The influence of using different backbones and the presence of a local transformer is tested. The results show that as long as there is local information (DGCNN / Point Transformer) present in the tokenization process, the performance remains competitive. On the other hand, using only PointNet or MLP for tokenization, the performance drops significantly.

| Backbone | PT | $EPE_{3D}\downarrow$ all | $ACC_S\uparrow$ all | $ACC_R\uparrow$ all | $Outliers\downarrow$ all | $EPE_{3D}\downarrow$ non-occ | $ACC_S\uparrow$ non-occ | $ACC_R\uparrow$ non-occ | $Outliers\downarrow$ non-occ |
|---|---|---|---|---|---|---|---|---|---|
| DGCNN | ✓ | **0.040** | **92.64** | **95.84** | 10.34 | **0.022** | **95.94** | **98.06** | 8.13 |
| DGCNN | | 0.052 | 89.68 | 94.37 | 13.71 | 0.030 | 93.74 | 97.14 | 11.00 |
| PointNet | ✓ | 0.043 | 92.22 | 95.80 | 10.86 | 0.024 | 95.65 | 98.04 | 8.63 |
| PointNet | | 0.063 | 86.76 | 93.06 | 16.67 | 0.037 | 91.45 | 96.31 | 13.51 |
| MLP | ✓ | 0.043 | 91.81 | 95.48 | **10.21** | 0.023 | 95.43 | 97.84 | **7.75** |
| MLP | | 0.060 | 88.08 | 93.33 | 14.11 | 0.035 | 92.69 | 96.55 | 10.83 |

Table 8: **Ablation study on the number of feature dimensions on F3D$_c$.** The performance decreases as the number of feature dimensions drops.

| dim | $EPE_{3D}\downarrow$ all | $ACC_S\uparrow$ all | $ACC_R\uparrow$ all | $Outliers\downarrow$ all | $EPE_{3D}\downarrow$ non-occ | $ACC_S\uparrow$ non-occ | $ACC_R\uparrow$ non-occ | $Outliers\downarrow$ non-occ |
|---|---|---|---|---|---|---|---|---|
| 32 | 0.073 | 83.04 | 91.32 | 21.07 | 0.044 | 88.36 | 95.07 | 17.56 |
| 64 | 0.051 | 89.64 | 94.57 | 13.68 | 0.029 | 93.79 | 97.32 | 10.93 |
| 128 | **0.040** | **92.64** | **95.84** | **10.34** | **0.022** | **95.94** | **98.06** | **8.13** |

represent the warped source point cloud toward the target point cloud. As we see in the figure, the blue points align very well with the green points, which demonstrates the effectiveness of our method.

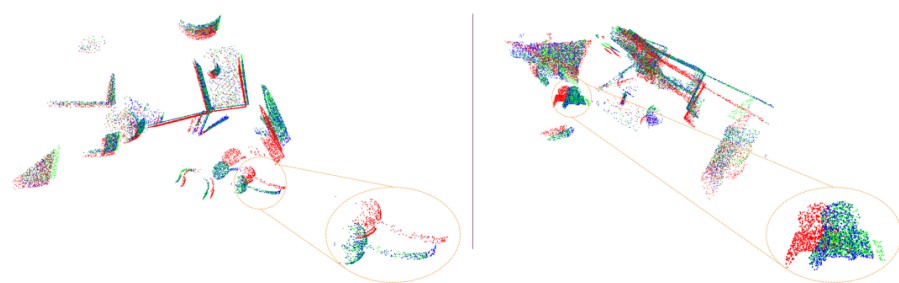

Figure 3: **Visualization results on FlyingThings3D.** Two scenes from the FlyingThings3D dataset are given. Red, blue, and green points represent the source, target, and warped source point cloud, respectively. Part of the point cloud is zoomed in for better visualization.

## 5  Conclusion

We propose to solve scene flow estimation from point clouds by a simple single-scale one-shot global matching, where we show that reliable feature similarity between point pairs is essential and sufficient to estimate accurate scene flow. To extract high-quality feature representations, we introduce a hybrid local-global-cross transformer architecture. Experiments show that both the presence of local information in the tokenization step and the stack of global-cross transformers are essential to success. GMSF shows state-of-the-art performance on the FlyingThings3D, KITTI Scene Flow, and Waymo-Open datasets, demonstrating the effectiveness of the method.

**Limitations:**  The global matching process in the proposed method needs to be supervised by the ground truth, which is difficult to obtain in the real world. As a result, most of the supervised scene flow estimations are trained on synthetic datasets. We plan to extend our work to unsupervised settings to exploit real data.

**Acknowledgements**: This work was partly supported by the Wallenberg Artificial Intelligence, Autonomous Systems and Software Program (WASP), funded by Knut and Alice Wallenberg Foundation, and the Swedish Research Council grant 2022-04266; and by the strategic research environment ELLIIT funded by the Swedish government. The computational resources were provided by the National Academic Infrastructure for Supercomputing in Sweden (NAISS) at C3SE partially funded by the Swedish Research Council grant 2022-06725, and by the Berzelius resource, provided by the Knut and Alice Wallenberg Foundation at the National Supercomputer Centre.

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
