# OpenReview forum: "GMSF: Global Matching Scene Flow"
_NeurIPS.cc/2023/Conference — NeurIPS 2023 poster_

### Official Review · Reviewer_vPo9 · 2023-06-26

**Soundness:** 3 good
**Presentation:** 3 good
**Contribution:** 3 good
**Rating:** 7
**Confidence:** 5

**Summary:**

Previous scene flow estimation methods require complicated coarse-to-fine or recurrent architectures as a multi-stage refinement. In contrast, this paper proposes a simpler single-scale one-shot global matching to address the problem. To this end, this paper decomposes the feature extraction step via a hybrid local-global-cross transformer architecture. Extensive experiments show that the proposed method achieves SOTA performance on multiple scene flow estimation benchmarks.

**Strengths:**

1.	This paper proposes Global Matching Scene Flow (GMSF) and achieves state-of-the-art performance on multiple scene flow estimation benchmarks.
2.	The proposed pipeline is simple and effective.
3.	The authors have provided the code in the submission.

**Weaknesses:**

1 Some related studies have been neglected. Please compare with the previous study [cite1] in experiments. Besides, previous studies [cite2-3] need to be cited and discussed.

[cite1] Lang I, Aiger D, Cole F, et al. SCOOP: Self-Supervised Correspondence and Optimization-Based Scene Flow[J]. arXiv e-prints, 2022: arXiv: 2211.14020.
[cite2] Li X, Kaesemodel Pontes J, Lucey S. Neural scene flow prior[J]. Advances in Neural Information Processing Systems, 2021, 34: 7838-7851.
[cite3] Li R., Zhang C., Lin G., Wang Z., Shen C.: Rigidflow: Self-supervised scene flow learning on point clouds by local rigidity prior. In Proceedings of the IEEE/CVF Conference on Computer Vision and Pattern Recognition (CVPR), IEEE Computer
Society, Los Alamitos, CA, USA (June 2022), pp. 16959–16968.

2 Compared methods are not consistent in Table 2 and Table 3. Please clarify the reason.

3 The motivation of Eq. (11) is unclear. Please further clarify the reason why this term can be viewed as a smoothing procedure.

4 Ablation study is not enough. I suggest the authors conduct an ablation study to Eq. (13). Specifically, two version models need to be compared, i.e., model A trained with the first term and model B trained with Eq. (13).

5 The authors need to compare the FLOPs, GPU memory, and run-time of recent scene flow methods. Although the proposed method is simple and effective, the computational cost needs to be compared.

6 It seems that tokenization is too complex and redundant. Specifically, Table 5 shows that DGCGG+PT achieves almost the same performance as MLP+PT. Therefore, I think only using PT is enough, and I suggest the authors conduct experiments to report the performance of GMSF with only PT as the tokenization process. In this way, GMSF would become more efficient.

7 Whenever any abbreviations appear for the first time, it requires a full form. For example, GMSF and LiDAR.


**Questions:**

Please see the Weaknesses. If these concerns are addressed, I am willing to improve the rating.

---

> ### Author Rebuttal · Authors · 2023-08-02
>
> We thank the reviewer for their positive comments and helpful suggestions. Below are our clarifications for the questions.
>
> **Q1: Related work.**
>
> We thank the reviewers for the suggestion of the unmentioned related work. We will add the following discussion in the related work.
>
> Different from the proposed method, which is fully supervised and trained offline, some other works focus on runtime optimization, prior assumptions, and self-supervision.
>
> Li $et\ al.$ [2] revisit the need for explicit regularization in supervised scene flow learning. The deep learning methods tend to rely on prior statistics learned during training, which are domain-specific. This does not guarantee generalization ability during testing. To this end, Li $et\ al.$ proposes to rely on runtime optimization with scene flow prior as strong regularization.
> Based on [2] Lang $et\ al.$ [1] propose to combine runtime optimization with self-supervision. A correspondence model is first trained to initialize the flow. Refinement is done by optimizing the flow refinement component during runtime. The whole process can be done under self-supervision.
> Pontes $et\ al.$ [4] use the graph Laplacian of a point cloud to force the scene flow to be "as rigid as possible". Same as Li $et\ al.$ [2], this constraint can be optimized during runtime.
> Li $et\ al.$ [3] propose a self-supervised scene flow learning approach with local rigidity prior assumption for real-world scenes. Instead of relying on point-wise similarities for scene flow estimation, region-wise rigid alignment is enforced.
>
> The comparison with SCOOP is provided here and will be added in the experiments section.
> |Method|$EPE_{3D}\downarrow$|$ACC_{S}\uparrow$|$ACC_{R}\uparrow$|$Outliers\downarrow$|
> |-|-|-|-|-|
> |SCOOP|0.063|79.7|91.0|24.4|
> |SCOOP+|0.047|91.3|95.0|18.6|
> |**GMSF**|0.051|79.9|92.3|22.9|
>
> SCOOP only reports results on KITTI$\_o$. The symbol $+$ indicates that all the points in the test point clouds are used during evaluation. Without the symbol indicates using the same dataset as us preprocessed by Liu $et\ al.$ in FlowNet3D [5].
> Different from our proposed method, which is fully trained offline and generalizes to KITTI without fine-tuning, SCOOP combines offline training with runtime optimization. The performance is increased by utilizing more data during evaluation (SCOOP$+$). Our proposed method does not require expensive online optimization and performs slightly worse in terms of $EPE_{3D}$, which may shed light on future work. Thanks for the suggestion.
>
> **Q2: Compared methods are not consistent in Table 2 and Table 3.**
>
> The datasets used in Table 2 and Table 3 are the same (Flyingthings3D and KITTI Scene Flow). However, the preprocessing procedures are different. We follow the preprocessing procedure from Liu $et\ al.$ in FlowNet3D [5] to prepare the F3D$\_o$/KITTI$\_o$ in Table 2, and follow the preprocessing procedure from Gu $et\ al.$ in HPLFlowNet [6] to prepare the F3D$\_s$/KITTI$\_s$ in Table 3. The difference is that F3D$\_o$/KITTI$\_o$ keeps all valid points with an occlusion mask available during training and testing, F3D$\_s$/KITTI$\_s$ simplifies the task by removing all occluded points (see paper L 234-240). The numbers are not consistent because of the different experimental settings.
>
> We take the results directly from the published papers. Some of the methods only conduct experiments on one version of the datasets. For better comparison with state-of-the-art methods, we conduct experiments with both of the settings.
>
> **Q3: Motivation of Eq. (11) and why it can be viewed as a smoothing procedure.**
>
> Please refer to the global response to all reviewers C.
>
> **Q4: Ablation study on Eq. (13).**
>
> Please refer to the global response to all reviewers D.
>
> **Q5: FLOPs, GPU memory, and Runtime.**
>
> Please refer to the global response to all reviewers A.
>
> **Q6: GMSF with only PT as the tokenization process.**
>
> Thanks for the suggestion. We interpret the suggestion as:
> Since employing DGCNN and MLP gives pretty much the same results, which leads to the question if further simplifying the architecture, e.g. by removing MLP and only employing PT, would work.
>
> This is not the case, since the purpose of DGCNN and MLP is to project the 3D coordinate into the high-dimensional feature space.
> The same procedure is employed in the PT [7] original paper, where the input feature vectors are acquired by applying MLP on the 3D coordinates.
> Removing the MLP and directly applying the PT on the 3-dimensional coordinate would result in a lack of capacity of the model.
> To support our claim that reducing the number of channels leads to a lack of capacity of the model, we did a small ablation study on the number of channels in the table below:
> |Channels|$EPE_{3D}\downarrow$|$ACC_{S}\uparrow$|$ACC_{R}\uparrow$|$Outliers\downarrow$|$EPE_{3D}\downarrow$|$ACC_{S}\uparrow$|$ACC_{R}\uparrow$|$Outliers\downarrow$|
> |-|-|-|-|-|-|-|-|-|
> ||$all$||||$nonocc$||||
> |64|0.059|87.06|93.43|17.13|0.032|92.64|96.98|13.57|
> |128|0.049|90.08|94.72|13.08|0.025|94.98|97.78|9.87|
>
> **Q7: Abbreviations.**
>
> Thanks for pointing out. We've made modifications based on the comment.
>
> **References**
>
> [1] Lang I, Aiger D, Cole F, et al. Scoop: Self-supervised correspondence and optimization-based scene flow[C]//CVPR2023.
>
> [2] Li X, Kaesemodel Pontes J, Lucey S. Neural scene flow prior[J]. NeurIPS2021.
>
> [3] Li R, Zhang C, Lin G, et al. Rigidflow: Self-supervised scene flow learning on point clouds by local rigidity prior[C]//CVPR2022.
>
> [4] Pontes J K, Hays J, Lucey S. Scene flow from point clouds with or without learning[C]//3DV2020.
>
> [5] Liu X, Qi C R, Guibas L J. Flownet3d: Learning scene flow in 3d point clouds[C]//CVPR2019.
>
> [6] Gu X, Wang Y, Wu C, et al. Hplflownet: Hierarchical permutohedral lattice flownet for scene flow estimation on large-scale point clouds[C]//CVPR2019.
>
> [7] Zhao H, Jiang L, Jia J, et al. Point transformer[C]//ICCV2021.

---

> > ### Comment · Reviewer_vPo9 · 2023-08-11
> >
> > Thank you for your response. I still have the following concerns about GMSF.
> >
> > (1) The comparison with SCOOP [cite1].
> > Because the authors have provided codes, I trained both SCOOP and GMSF by myself. However, SCOOP is faster, data-efficient (one-tenth of the whole training set), and easy to be trained (less than 1h with a single 3090 trained on FT3D). In contrast, the GPU requirement of GMSF is too large, and I cannot train the GMSF on a GeForce RTX 3090 (24G) even with bs = 1. More crucially, SCOOP is an unsupervised method without ground truth flows. Please clarify the strength of GMSF.
> >
> > (2) About the tokenization process.
> > I am confused about the table in the answer to Q6. Which row of the table represents the performance of GMSF with only PT as the tokenization process? According to Concern (1), I strongly suggest the authors simplify the architecture of GMSF. Otherwise, it is impossible to use  GMSF for real-world applications.

---

> > > ### Author Response · Authors · 2023-08-11
> > > **Thanks for the reply and for the effort of running the experiments.**
> > >
> > > Thanks for the reply and for the effort of running the experiments.
> > >
> > > Q1: The strength of GMSF.
> > >
> > > Methods, such as SCOOP, based on prior information and runtime optimization are usually easier to train but more time-consuming during inference ($10\times$–$100\times$ slower, refer to [1]). Therefore, when there is a requirement on inference time, runtime optimization-based methods are not suitable. For example, SCOOP [2] achieves the best performance with an inference time of \~20s, Neural Prior [1] with an inference time of \~40s (see Figure 7. in SCOOP). This is around two orders of magnitude higher than our inference time (371.8ms).
> > > Moreover, during inference, our method requires 5GB of memory, and can thus run on a Geforce RTX 3090 or lower end graphics cards.
> > >
> > > Q2: Table in the answer to Q6.
> > >
> > > The result of GMSF with only PT is not in the table.
> > > The table is with DGCNN+PT, but the dimensions of the feature space are reduced from 128 to 64 in the first row.
> > > As mentioned, the function of DGCNN and MLP is to project the 3D coordinate into the high-dimensional feature space. Without them and only applying PT on the 3D coordinate would result in a lack of capacity of the model.
> > > The table shows a performance drop with fewer feature dimensions, and presumably a further simplification could result in a larger performance drop.
> > > (We have tried training GMSF with only PT, but this has problems with convergence.)
> > >
> > > However, another way to simplify our model is to reduce the number of global-cross transformer layers (see Table 4 in the paper). The table shows that our proposed method has a good performance even with only 4 global-cross transformer layers.
> > > (We have tried with global-cross transformer layers = 4, batch size = 1. It converges well with 14.5GB memory usage.)
> > >
> > > Besides, since most of the applicability in real scenarios is based on inference, this makes inference time arguably the most important factor in the real-world applicability of a method. We are on par with the other state-of-the-art methods (**two orders of magnitude faster than SCOOP**). Moreover, during inference, our method only requires 5GB of memory, therefore, it can even be run on lower end consumer graphics cards. We hope that could help with real-world applications.
> > >
> > > References
> > >
> > > [1] Li X, Kaesemodel Pontes J, Lucey S. Neural scene flow prior[J]. NeurIPS2021.
> > >
> > > [2] Lang I, Aiger D, Cole F, et al. Scoop: Self-supervised correspondence and optimization-based scene flow[C]//CVPR2023.

---

> > > > ### Comment · Reviewer_vPo9 · 2023-08-12
> > > >
> > > > Thank you for your response. I partially agree with the authors' opinion that inference time may be the most important factor in real-world applications. Human annotations/labels requirements, generalization to out-of-distribution data, and computational complexity are all important factors. However, it is impossible for one paper to solve all the problems. In this way, I suggest the authors cite and discuss the recent paper [cite1] in the revision (no need to do this during the discussion phase).
> > > >
> > > > GMSF is effective and achieves SOTA performances among supervised scene flow methods. Although GMSF still has some drawbacks, such as high training difficulty and the supervised scheme, the contribution and novelty of GMSF are strong enough to be accepted. I think the added experiments should be included in the revision to further strengthen the paper. I am willing to raise my rating.
> > > >
> > > >
> > > > [cite1] Chodosh N, Ramanan D, Lucey S. Re-Evaluating LiDAR Scene Flow for Autonomous Driving[J]. arXiv preprint arXiv:2304.02150, 2023.

---

### Official Review · Reviewer_UZxB · 2023-07-03

**Soundness:** 3 good
**Presentation:** 3 good
**Contribution:** 2 fair
**Rating:** 5
**Confidence:** 4

**Summary:**

The paper presents GMSF, a transformer-based method that matches dense features to estimate the scene flow from point clouds. The proposed method uses a transformer-based architecture that matches two point clouds and calculates the scene flow leveraging the cross- and self-attention modules. The paper presents experiments on FlyingThings3D and KITTI Scene Flow datasets where the proposed method consistently improves the benchmarks.

**Strengths:**

In sum, I think the solution using cross- and self-attention mechanisms are an interesting application to solve the scene-flow problem. Although, this solution has been applied to sparse feature matching (see LoFTR), I think it is also good to see it works for the scene-flow problem. Here are the details of the strengths:

S1. I find the solution an interesting application of cross- and self-attention mechanisms to solve the scene flow problem. Though in general I think the novelty is not that high since the overall idea has been used for feature matching in LoFTR CVPR 2021 [36].

S2. The clarity of the narrative is quite good. The description of the architecture and attention modules is clear. Also, the description of the losses make sense and are easy to understand. Thus, I think the paper can be reproducible.

**Weaknesses:**

Overall, I think while the paper shows another application of attention to compute 3D scene flow, I think the paper is lacking more thorough experiments and ablations. Here are the details:

W1. Missing ablations. First, the paper is not showing the performance impact of the parameters of the KNN component shown in Figure 2.I am sure this is also crucial since this allows the encoder to grab local information. How to set the KNN parameters? What is the effect of this parameter in the final performance? Second. What is the behavior of setting $\lambda$ to a different value? Why was $\lambda=0.9$ and what is the performance of the method when varying this parameter?

W2. Is the KITTI Scene Flow dataset challenging enough? Since this is a dataset for autonomous driving, the motion of the vehicle is mainly planar and thus limiting the motion degrees of freedoms (only 1 for rotation, and mostly one 1 for translations, since the car moves linearly most of the time). I think this greatly simplifies the complexity of finding correspondence of any type (e.g., scene flow) in these autonomous driving datasets.

W3. Lastly, I am concerned about the novelty of the approach. I think previous works have shown that attention mechanisms are useful for matching tasks in vision, and thus I struggle to find novel components or ideas in the paper. I think the paper should discuss more in depth the novelties of the paper more in detail.

----
Post Rebuttal

After reading the rebuttal and discussion with the authors, my concerns have been addressed and I will increase my rating.

**Questions:**

1. As stated in Weaknesses section, how to set the KNN parameters of the method?
2. in line 158, the paper mentions "stable tokens". However, the paper never defines what a "stable token" is.

**Limitations:**

I think limitations are stated adequately.

---

> ### Author Rebuttal · Authors · 2023-08-03
>
> We thank the reviewer for their positive comments and helpful suggestions. Below are our clarifications for the questions.
>
> **Q1a: Ablation study of the KNN component.**
>
> For the Point Transformer in Figure 2, we follow the setting in Zhao $et al.$ [1] and set $k$ to 16. Thanks to the reviewer's suggestion to provide an ablation study on the $k$ parameter. The results are given in the following table:
>
> | $k$ | $EPE_{3D}\downarrow$ | $ACC_{S}\uparrow$ | $ACC_{R}\uparrow$ | $Outliers\downarrow$ | $EPE_{3D}\downarrow$ |$ACC_{S}\uparrow$ | $ACC_{R}\uparrow$ | $Outliers\downarrow$ |
> |-----------|-----------|-----------|-----------|-----------|-----------|-----------|-----------|-----------|
> |  | $all$ | $all$ | $all$ | $all$ | $nonocc$ | $nonocc$ | $nonocc$ | $nonocc$ |
> | $k=8$ | 0.050 | 89.06 | 94.25 | 14.33 | 0.027 | 94.25 | 97.50 | 10.94 |
> | $k=16$ | 0.049 | 90.08 | 94.72 | 13.08 | 0.025 | 94.98 | 97.78 | 9.87 |
> | $k=32$ | 0.049 | 89.53 | 94.42 | 13.89 | 0.026 | 94.61 | 97.58 | 10.59 |
>
> The result confirms that the best performance is obtained for $k=16$ in the Point Transformer paper [1] (Section 4.4 Ablation Study, Number of neighbors). Setting $k$ to 8 and 32 results in a slightly worse performance.
>
> **Q1b: Ablation study for the parameter $\gamma$ in Eq. (13).**
>
> We test the two extreme cases: with only the first term $\hat{V}\_{final}$, and only the second term $\hat{V}\_{inter}$, respectively.
> Thanks to the recommendation we could slightly improve the performance by simplifying Eq. (13) that was previously introduced in RAFT [2].
> Please refer to the global response to all reviewers D for more details.
>
> **Q2: KITTI Scene Flow dataset not challenging enough.**
>
> In the autonomous driving scenarios, the vehicles are mostly moving in a plane. However, the scene flow is often estimated from the view of a moving vehicle instead of a static view. The largest camera motion is yawing, but there is also a rotation, though smaller, around the other two axes. Therefore, the degrees of freedom of the relative movement of the other vehicles is not restricted to 1 translation and 1 rotation. Moreover, pedestrians, which are also important in autonomous driving scenarios, are much more complex since they are non-rigid objects.
>
> Although such autonomous driving scenarios are sometimes simplified by assuming the moving vehicles are rigid objects, explicitly modeling this fact oversimplifies the task and introduces a strong and potentially misleading motion prior. By contrast, our method does not exploit any such biases or prior information, thus being applicable to more general cases, as underlined by the good performance of our approach.
>
> Furthermore, we complement KITTI with the more challenging datasets, FT3D (see paper Table 2, 3) and Waymo (see response to reviewer QN7Z, Q1).
> FT3D has more complicated and less constrained object movements and is larger than the KITTI dataset. The Waymo dataset is another larger autonomous driving scene flow benchmark.
> We achieve superior performance on both datasets compared to previous methods.
>
> **Q3: Novelty.**
>
> Cross- and self-attention mechanisms proved efficient in 2D image matching [3].
> However, their application in the task of scene flow estimation has not been thoroughly studied yet.
> We propose a "simple and effective" (vPo9) architecture to tackle scene flow estimation, which is proven effective on multiple scene flow benchmarks, including the new challenging Waymo-open dataset (cf. response to Q1 to QN7Z).
> Moreover, the novelty and contribution of our work is not only the architecture itself but also the evidence that it is applicable and works very well in a new field.
> Similarly, Reviewer D7Yn commented that the proposed architecture, and the motivation behind it, are intuitive and novel, and its application to the task of scene flow seems conceptually new.
>
> **Q4: Definition of a "stable token".**
>
> We thank the reviewer to point out our inaccurate wording. "Stable token" indicates that the token with the Local Point Transformer is more powerful and helps with both accuracy and robustness.
>
> **References**
>
> [1] Zhao H, Jiang L, Jia J, et al. Point transformer[C]//ICCV2021.
>
> [2] Teed Z, Deng J. Raft: Recurrent all-pairs field transforms for optical flow[C]//ECCV2020.
>
> [3] Sun J, Shen Z, Wang Y, et al. LoFTR: Detector-free local feature matching with transformers[C]//CVPR2021.

---

> > ### Comment · Reviewer_UZxB · 2023-08-15
> > **RE: Rebuttal by Authors**
> >
> > Thanks for the clarifications. A few more questions:
> >
> > Q1a: It seems like the parameter $k$ has marginal impact, and that k=16 "best" performance may be within the margin of error. Any insight as to why $k$ has a marginal impact on the method?
> >
> > Q2: The reason I raised this concern is to understand the generalization of the proposed method. Although the results from FT3D partially satisfy my question, the FT3D dataset is synthetic and may pose a synthetic-real gap.

---

> > > ### Author Response · Authors · 2023-08-15
> > >
> > > Thanks for the reply.
> > >
> > > Q1: Marginal impact of $k$.
> > >
> > > We believe that the marginal impact of $k$ is due to the design choice of our tokenization process. The tokenization consists of a DGCNN and a Point Transformer, both of which encode local information. Table 5 in the paper shows that as long as there is some local information encoded, the performance remains competitive (EPE=0.049 for DGCNN+PT, EPE=0.055 for DGCNN). Only changing the $k$ parameter in the Point Transformer does not affect the local information encoded by DGCNN, thus has a limited impact.
> > >
> > > Moreover, Local information of only 4 points already includes features such as curvature. These features become more distinct the more neighboring points are considered. But there is a natural limit when they don't add any more information.
> > >
> > > Q2: Performance on the real dataset.
> > >
> > > Since the annotation of real data is very expensive in scene flow estimation, synthetic datasets are usually employed during training [1, 2]. This is also the case for optical flow estimation [3, 4], where the same FT3D dataset is used during training and shows a good generalization performance on the KITTI dataset.
> > >
> > > However, we agree that when training only on synthetic datasets, the synthetic-to-real-gap should be carefully analyzed.
> > > According to the suggestion from Reviewer QN7Z, we extend our experiments to the larger and more challenging Waymo-open autonomous driving dataset that contains 798 training and 202 validation sequences (each 20 seconds with 10Hz point clouds data) with more complex scenes compared to the KITTI dataset.
> > >
> > > The results in comparison with state-of-the-art methods are given in the following table. We outperform the state of the art by a large margin. We hope this could help with your concern about the applicability of our method in real-world scenarios.
> > >
> > > | method | $EPE_{3D}\downarrow$ | $ACC_{S}\uparrow$ | $ACC_{R}\uparrow$ | $Outliers\downarrow$ |
> > > |-----------|-----------|-----------|-----------|-----------|
> > > | FlowNet3D[1] | 0.225 | 23.0 | 48.6 | 77.9 |
> > > | PointPWC[5] | 0.307 | 10.3 | 23.1 | 78.6 |
> > > | FESTA[6] | 0.223 | 24.5 | 27.2 | 76.5 |
> > > | FH-Net[7] | 0.175 | 35.8 | 67.4 | 60.3 |
> > > | **GMSF** | 0.086 | 73.9 | 84.7 | 43.9 |
> > >
> > > References:
> > >
> > > [1] Liu X, Qi C R, Guibas L J. Flownet3d: Learning scene flow in 3d point clouds[C]//CVPR2019.
> > >
> > > [2] Gu X, Wang Y, Wu C, et al. Hplflownet: Hierarchical permutohedral lattice flownet for scene flow estimation on large-scale point clouds[C]//CVPR2019.
> > >
> > > [3] Teed Z, Deng J. Raft: Recurrent all-pairs field transforms for optical flow[C]//ECCV2020.
> > >
> > > [4] Jiang S, Campbell D, Lu Y, et al. Learning to estimate hidden motions with global motion aggregation[C]//ICCV2021.
> > >
> > > [5] Wu W, Wang Z Y, Li Z, et al. Pointpwc-net: Cost volume on point clouds for (self-) supervised scene flow estimation[C]//ECCV2020.
> > >
> > > [6] Wang H, Pang J, Lodhi M A, et al. Festa: Flow estimation via spatial-temporal attention for scene point clouds[C]//CVPR2021.
> > >
> > > [7] Ding L, Dong S, Xu T, et al. Fh-net: A fast hierarchical network for scene flow estimation on real-world point clouds[C]//ECCV2022.

---

> > > > ### Comment · Reviewer_UZxB · 2023-08-19
> > > >
> > > > Thanks for answering my concerns I will increase my rating.

---

### Official Review · Reviewer_YAmp · 2023-07-04

**Soundness:** 3 good
**Presentation:** 3 good
**Contribution:** 2 fair
**Rating:** 5
**Confidence:** 4

**Summary:**

This work aims to address the task of scene flow estimation for 3D point clouds. The authors propose a hybrid architecture based on local-global-cross transformers. Given as input a source and a target point cloud, first, the local transformer extracts geometric features within a patch, then the global transformer analyzes each point cloud individually using self-attention to capture the overall context, and finally the cross transformer exchanges information between the point clouds to generate the final feature representation for each point. The scene flow is predicted by performing pointwise matching with the cross similarity matrix, while occlusions are handled through a self-similarity matrix applied to the predicted scene flow. To evaluate the effectiveness of their approach, the authors conduct experiments on two benchmarks for scene flow estimation, demonstrating better performance compared to existing methods.

**Strengths:**

-  This work introduces a straightforward architecture for scene flow estimation, utilizing transformers. Without bells and whistles, the authors show that the proposed network can produce discriminative per-point features that can be robustly matched for scene flow computation.
-  The authors conduct extensive experiments on the FlyingThings3D and KITTI Scene Flow benchmarks in different preprocessing and occlusion settings. The results suggest that the proposed method has significant performance improvement on FlyingThings3D, while also performing competitively on KITTI Scene Flow when compared to existing methods for generalization test.
-  The paper is overall well written and easy to follow.

**Weaknesses:**

-  One concern for this work is its limited technical contributions: the hybrid network uses the well-established transformer architecture, while the scene flow is estimated by a common probabilistic point matching approach. One seemingly interesting proposal is the occlusion handling with the self-similarity matrix. However, it lacks in-depth explanations for why it helps with occlusion handling, and its ablation study is also missing in Sec. 4.5.
-  In the generalization test on KITTI-S (Tab. 3), the proposed method exhibits a performance gap compared to PT-FlowNet [8]. More detailed analysis and explanation would be helpful for gaining a better understanding of this discrepancy.
- For the ablation study in Tab. 4, it is unclear whether the performance saturates with eight global-cross transformer layers, and whether more layers would be beneficial or not. To provide a comprehensive assessment, it would be good to include comparisons of runtime and memory usage, as the backbone is built upon transformers, which may not scale well with more points, and the paper mentions that input point clouds are resampled to 8K points.
-  Minor: L267, “Although we don’t” => Note that we do not


**Questions:**

Please see the Weaknesses section.


**Limitations:**

The authors discussed limitations at the end of the paper.

---

> ### Author Rebuttal · Authors · 2023-08-04
>
> We thank the reviewer for their positive comments and helpful suggestions. Below are our clarifications for the questions.
>
> **Q1a: Limited technical contributions.**
>
> The transformer architecture and probabilistic matching approach have been studied and proven efficient previously in many NLP tasks and image understanding.
> However, their application in the task of scene flow hasn't been thoroughly studied.
> We propose a "simple and effective" (vPo9) architecture to tackle scene flow estimation, which is proven effective on multiple scene flow benchmarks, including the new challenging Waymo-open dataset (cf. response to Q1 to QN7Z).
> Moreover, the novelty and contribution of our work is not only the architecture itself but also the evidence that it is applicable and works very well in a new field.
> Similarly, Reviewer D7Yn commented that the proposed architecture, and the motivation behind it, are intuitive and novel, and its application to the task of scene flow seems conceptually new.
>
> **Q1b: Why occlusion handling works and can be viewed as a smoothing procedure.**
>
> Please refer to the global response to all reviewers C.
>
> **Q1c: Ablation study on occlusion handling.**
>
> Please refer to the global response to all reviewers D.
>
> **Q2: Performance gap compared to PT-FlowNet.**
>
> We would like to first mention that PT-FlowNet was officially published on May 2023 with a preliminary online version available already in March (the NeurIPS submission deadline was on May 11th). For fairness and comprehensiveness we chose to include the paper and compare it with our proposed method.
>
> The proposed method shows a significantly better performance on FlyingThings3D dataset compared to PT-FlowNet, especially in terms of $ACC_S$ and $Outlier$ (accuracy: 98.9\% vs. 91.4\%; outlier: 17.4\% vs. 3.2\%).
> When generalizing to the KITTI dataset, the proposed method exhibits a performance gap in terms of $EPE_{3D}$. However, the $ACC_R$ is on par with PT-FlowNet, and the other metrics only differ slightly. Besides, KITTI is a relatively small dataset with only less than 200 scenes for evaluation.
>
> One aspect that could explain our performance gap compared to PT-FlowNet is that our proposed method is not specifically designed for the experimental setting without occlusion.
> In the real-world scenario, there is never the case where we can get clean, non-occluded point pairs. Therefore, we aim for more general cases where occlusion exists. The generalization ability on KITTI$_o$ (see paper Table 2.) well supports our claim.
> However, PT-FlowNet only provides results on F3D$_s$ and KITTI$_s$ datasets.
>
> Moreover, during testing, PT-FlowNet requires 32 refinement iterations which slows down the runtime. For each of the scenes, PT-FlowNet needs 898.5 ms to process, while our method only needs 371.8 ms, more than twice as fast. Increasing the number of attention blocks in our model likely results in better performance at the cost of speed (see additional experiments on the number of transformer layers in the global response to all reviewers B.).
>
> In summary: except for the only partially realistic KITTI$_s$ setting, where we almost match PT-FlowNet's performance, outperform them by a large margin while at the same time requiring lower computational resources.
>
> **Q3a: Ablation study on the number of transformer layers.**
>
> Please refer to the global response to all reviewers B.
>
> **Q3b: FLOPs, GPU memory, and Runtime.**
>
> Please refer to the global response to all reviewers A.
>
> **Q3c: Point clouds are resampled to 8K points.**
>
> For clarification, this is a common setting for scene flow estimation from point clouds. We follow the same setting for a fair comparison.
>
> **Q4: Reformulation.**
>
> We thank the reviewer for the suggested reformulation and will use it in the final version.

---

> > ### Comment · Reviewer_YAmp · 2023-08-18
> > **Comments on Authors' Rebuttal**
> >
> > I appreciate the authors' effort in addressing my questions in the rebuttal. After reading through the review comments from other reviewers, although I am still concerned about the technical novelty, as commented by the fellow reviewers, I think this work can serve a stepping stone for future works on scene flow given its comprehensive and improved results. It would be great if the authors can release the code and trained models to benefit the community. As of now, I would like to keep my rating.

---

### Official Review · Reviewer_QN7Z · 2023-07-07

**Soundness:** 2 fair
**Presentation:** 3 good
**Contribution:** 2 fair
**Rating:** 5
**Confidence:** 5

**Summary:**

This paper proposes a hybrid local-global-cross transformer scene flow estimation model, achieving the state-of-the-art results on FlyingThings3D and KITTI Scene Flow datasets.

**Strengths:**

1. This work has a mature model design based on Transformers, achieving superior results on the Flyingthings3D and KITTI Scene flow datasets.

2. The structure and writing of this paper do not have major issues.

**Weaknesses:**

1. Scene flow estimation has been developed for many years, and the autonomous driving industry has already introduced Occupancy and Flow Prediction techniques. Researchers should not be limited to designing a toy model solely to maximize the benchmarks on FlyingThings3D and KITTI Scene Flow. These two datasets have dense point clouds and clear correspondences, which are far from practical applications. For this paper, I hope the authors can conduct more experiments on the Waymo motion data (https://waymo.com/open/data/motion/).

2. Undoubtedly, Transformers will bring more powerful feature modeling capabilities and higher latency. How does the latency of this method compare to competitors? Please add this item to the main experimental table.

**Questions:**

Please refer to the weaknesses section, refute views or answer questions or improve the paper.

**Limitations:**

The limitations has been discussed.

---

> ### Author Rebuttal · Authors · 2023-08-04
>
> Thanks for the comments and suggestions. We improve the paper based on the two weaknesses pointed out. We show how we improve the paper below:
>
> **Q1: Experiments on the Waymo-open dataset.**
>
> As suggested, we conduct additional experiments on the Waymo-open dataset for scene flow.
> The dataset contains 798 training and 202 validation sequences. Each sequence consists of 20 seconds of 10Hz point cloud data.
> Due to the large scale of the dataset and the time limitation, we trained our model on 1/8 of the training sequences (first 100 training sequences) and tested on all 202 validation sequences.
> We use the same parameter setting and training scheme as training on FlyingThings3D, and train for 300k iterations.
> The results in comparison with other state-of-the-art methods are given in the following table:
>
> | method | $EPE_{3D}\downarrow$ | $ACC_{S}\uparrow$ | $ACC_{R}\uparrow$ | $Outliers\downarrow$ |
> |-----------|-----------|-----------|-----------|-----------|
> | FlowNet3D | 0.225 | 23.0 | 48.6 | 77.9 |
> | PointPWC | 0.307 | 10.3 | 23.1 | 78.6 |
> | FESTA | 0.223 | 24.5 | 27.2 | 76.5 |
> | FH-Net | 0.175 | 35.8 | 67.4 | 60.3 |
> | **GMSF** | 0.086 | 73.9 | 84.7 | 43.9 |
>
> Our proposed method already demonstrates a massive improvement over all the other methods when only trained on 1/8 of the training set. We are convinced that training on the full dataset would result in an even better performance. The results of the full training will be added to the main paper.
>
> **Q2: FLOPs, GPU memory, and Runtime.**
>
> Please refer to the global response to all reviewers A.
>
> Since the additional results on the Waymo dataset show promising performance and all our reported memory and runtime studies are on par with previous approaches while significantly improving the performance, we hope to have fully answered all questions the reviewer has. We are open to elaborate on these points further during the discussion phase.

---

> > ### Comment · Area_Chair_72D6 · 2023-08-19
> > **Has the rebuttal addressed your concerns**
> >
> > Dear Reviewer QN7Z,
> >
> > Could you please read the author rebuttal and acknowledge if your concerns have been addressed? The discussion period will end very soon on Monday, August 21. Thank you for your time in reviewing this submission!
> >
> > Best,
> >
> > AC

---

> > ### Comment · Reviewer_QN7Z · 2023-08-21
> >
> > Sorry for my late reply. If the method can bring such a significant improvement, I will change the score to positive. Please merge the new waymo experiment into the camera ready version. Thanks.

---

### Official Review · Reviewer_D7Yn · 2023-07-07

**Soundness:** 3 good
**Presentation:** 3 good
**Contribution:** 2 fair
**Rating:** 5
**Confidence:** 5

**Summary:**

The paper proposes GMSF for scene flow estimation from point clouds.
As far as the authors are aware, GMSF is the first to address scene flow estimation with global matching - GMSF is formulated as a single-scale one-shot global matching.
GMSF incorporates a novel local-global-cross transformer architecture to extract high-quality feature representation, to finally compute the scene flow between point clouds via global matching.
GMSF outperforms existing methods on F3D_c, F3D_o, F3D_s and KITTI_o, while performing competitively on KITTI_s.
The ablative results show that increasing the number of global-cross transformer layers - thus increasing the capacity - is beneficial to the performance, and that the presence of local information is crucial for the performance of GMSF.

**Strengths:**

* The paper is overall well written and easy to follow.

* The proposed architecture, and the motivation behind it, is intuitive and novel. While the architectural novelty itself is not entirely new (e.g., the sequence of local and global attention has first been introduced by SuperGlue[1] and LoFTR[2] for 2D image matching, and has been applied to 3D point cloud registration through methods including CoFiNet[3] and GeoTransformer[4]), its application to the task of scene flow seems conceptually new.

* Strong performances on the standard benchmarks of scene flow estimation.

[1] PE Sarlin et al., SuperGlue: Learning Feature Matching with Graph Neural Networks, CVPR 2020
[2] J Sun et al., LoFTR: Detector-Free Local Feature Matching with Transformers, CVPR 2021
[3] H Yu et al., CoFiNet: Reliable Coarse-to-fine Correspondences for Robust Point Cloud Registration, NeurIPS 2021
[4] Z Qin et al., Geometric Transformer for Fast and Robust Point Cloud Registration, CVPR 2022

**Weaknesses:**

* Insufficient ablative experiments. What if the number of transformer layers increase to a number higher than 8? The given results show that the performance improves gracefully with the number of layers, and it naturally leaves the question to 'until how much'. Also, what if the global transformer and cross transformers are decoupled, such that they can have varying number of layers?

* Lack of latency and computation analyses. The authors emphasize that GMSF is a single-scale, one-shot method; then how does it compare to existing methods in terms of latency and computation (FLOPs, memory)?

* Lack of analysis. How does incorporating global-cross transformer layers improve the performance, and how does incorporating **more** layers **further** improve the performance? This has been partially answered by Table 4, but a visual comparison / analysis would be more convincing.

* Lack of mention of 3D point cloud registration methods. While the task at hand is different, the architectural design and motivation is closely related to 3D point cloud registration, which I believe is therefore worth mentioning in the related work section. Also, it might be a bit of an overstatement to mention that GMSF is the 'first' to address scene flow estimation with global matching, as scene flow estimation and point cloud registration are seemingly closely related tasks .

**Questions:**

Please refer to the weaknesses section. The motivation and the proposed method are intuitive and novel, but the design choice of GMSF should be better substantiated, with included analyses for clarity.

**Limitations:**

The authors have addressed the limitations of GMSF in the paper.

---

> ### Author Rebuttal · Authors · 2023-08-04
>
> We thank the reviewer for their positive comments and helpful suggestions. Below are our clarifications for the questions.
>
> **Q1a: Ablation study on the number of transformer layers.**
>
> Please refer to the global response to all reviewers B.
>
> **Q1b: Ablation study on decoupling global transformers and cross transformers.**
>
> We follow SuperGlue [1] and LoFTR [2] and do not decouple the global-cross transformer layer since this combination has been proven effective.
> We now add an ablation study on decoupling the transformer layer. Three architectures are compared:
> | Model | $EPE_{3D}\downarrow$ | $ACC_{S}\uparrow$ | $ACC_{R}\uparrow$ | $Outliers\downarrow$ | $EPE_{3D}\downarrow$ |$ACC_{S}\uparrow$ | $ACC_{R}\uparrow$ | $Outliers\downarrow$ |
> |-----------|-----------|-----------|-----------|-----------|-----------|-----------|-----------|-----------|
> |  | $all$ |||| $nonocc$ ||||
> | global | 0.141 | 64.65 | 81.17 | 40.89 | 0.069 | 72.11 | 88.52 | 34.84 |
> | cross | 0.051 | 88.65 | 94.04 | 14.73 | 0.027 | 93.97 | 97.41 | 11.20 |
> | global-cross | 0.049 | 90.08 | 94.72 | 13.08 | 0.025 | 94.98 | 97.78 | 9.87 |
>
> The result shows that the cross transformer layers significantly contribute to the performance. This coincides with the finding in SuperGlue [1] where they address "cross-attention is critical to strong gluing" in Section 5.4 and Table 4.
>
> **Q2: FLOPs, GPU memory, and Runtime.**
>
> Please refer to the global response to all reviewers A.
>
> **Q3: Visual comparison / analysis of the influence of incorporating more transformer layers.**
>
> We provide an additional visual analysis of our model with 0, 2, and 24 transformer layers in the attached PDF to show how incorporating global-cross transformer layers improves performance and how incorporating more layers further improves performance.
>
> The results show that without global-cross transformer layers the prediction performance drops when there's a lack of texture information e.g. planar and curved surfaces, repetitive pattern. By incorporating only 2 transformer layers, the performance is significantly improved, which is also shown in the paper in Table 4.
> Without global-cross transformer layers, the learned features are local features. Thus, when there is a planar or curved surface or a repetitive pattern, the matching process tends to be inaccurate.
> More transformer layers further help with such cases. However, this might not be easy to see in the visualization since the difference of the $EPE_{3D}$ is small (0.081m and 0.041m for transformer layers of 2 and 24, respectively).
>
> **Q4: Related work in 3D point cloud registration.**
>
> We agree that point cloud estimation is a closely related topic to scene flow estimation. We will discuss this relationship in the paper, also with regard to our contributions, and add it to the related work section. A brief summary is provided below:
>
> Related to scene flow estimation, there are some correspondence-based point cloud registration methods. Such methods separate the point cloud registration task into two stages: finding the correspondences and recovering the transformation.
>
> PPFNet [3]  and PPF-FoldNet [4]  proposed by Deng $\textit{et al.}$ focus on finding sparse corresponding 3D local features.
> Gojcic $\textit{et al.}$ [5] propose to use voxelized smoothed density value (SDV) representation to match
> 3D point clouds.
> These methods only compute sparse correspondences and are not capable of handling dense correspondences required in scene flow tasks.
> More related works are CoFiNet [6] and GeoTransformer [7], both of which involve finding dense correspondences employing transformer architectures.
> Yu $\textit{et al.}$ in CoFiNet [6] proposes a detection-free learning framework and find dense point correspondence in a coarse-to-fine manner. Qin $\textit{et al.}$ in GeoTransformer [7] further improves the accuracy by leveraging the geometric information.
> Some works further introduced superpoint matching to solve the ambiguity introduced by self-attention layers.
> RoITr [8] introduces a Rotation-Invariant Transformer to disentangle the geometry and poses, and tackle point cloud matching under arbitrary pose variations.
> PEAL [9] introduced prior embedded Explicit Attention Learning model (PEAL), and for the first time explicitly inject overlap prior into Transformer to solve point cloud registration under low overlap.
> However, the goal of point cloud registration is not to estimate the translation vectors for each of the points, which makes our work different from these approaches.
>
> **References**
>
> [1] Sarlin P E, DeTone D, Malisiewicz T, et al. Superglue: Learning feature matching with graph neural networks[C]//CVPR2020.
>
> [2] Sun J, Shen Z, Wang Y, et al. LoFTR: Detector-free local feature matching with transformers[C]//CVPR2021.
>
> [3] Deng H, Birdal T, Ilic S. Ppfnet: Global context aware local features for robust 3d point matching[C]//CVPR2018.
>
> [4] Deng H, Birdal T, Ilic S. Ppf-foldnet: Unsupervised learning of rotation invariant 3d local descriptors[C]//ECCV2018.
>
> [5] Gojcic Z, Zhou C, Wegner J D, et al. The perfect match: 3d point cloud matching with smoothed densities[C]//CVPR2019.
>
> [6] Yu H, Li F, Saleh M, et al. Cofinet: Reliable coarse-to-fine correspondences for robust pointcloud registration[J]. NeurIPS2021.
>
> [7] Qin Z, Yu H, Wang C, et al. Geometric transformer for fast and robust point cloud registration[C]//CVPR2022.
>
> [8] Yu H, Qin Z, Hou J, et al. Rotation-invariant transformer for point cloud matching[C]//CVPR2023.
>
> [9] Yu J, Ren L, Zhang Y, et al. PEAL: Prior-Embedded Explicit Attention Learning for Low-Overlap Point Cloud Registration[C]//CVPR2023.

---

> > ### Comment · Reviewer_D7Yn · 2023-08-19
> >
> > Thank you authors for the detailed responses to my concerns. They seem to have been well addressed. I still appreciate the conceptual novelty and the strong performance of the paper, but am still concerned about the limited technical novelty of the paper. I would like to maintain my current rating of borderline accept.

---

### Author Rebuttal · Authors · 2023-08-03

We thank the reviewers for their valuable feedback and questions.
We are happy that the reviewers find our proposed approach, and the motivation "intuitive and novel" (D7Yn).
The architecture is "straightforward without bells and whistles" (YAmp), "mature" (QN7Z), and "simple and effective" (vPo9).
The solution is an "interesting application" of cross- and self-attention mechanisms to solve the scene flow problem (UZxB).
The paper is well-written and easy to follow (D7Yn, YAmp, UZxB).
The strong performance on the standard benchmarks of scene flow estimation is highlighted by (D7Yn, QN7Z, YAmp, vPo9).

In the following, we will address the questions asked by multiple reviewers. Individual questions are addressed below the respective reviews.
The provided additional results are mostly ablation studies suggested by the reviewers that complement our existing experiments and provide a deeper insight into our approach.

**A: FLOPs, GPU memory, and Runtime (D7Yn, QN7Z, YAmp, vPo9).**

Several reviewers requested meta-information on our experiments.
Adding to our ablation results, we report runtime (ms per scene) during testing on an NVIDIA A40 GPU, FLOPs (G), Number of parameters (M), GPU memory (GB) during testing (batch size 1) and training (batch size 8) under different numbers of transformer layers in the following table:
|layers|0|2|4|6|8|10|12|16|24|
|-|-|-|-|-|-|-|-|-|-|
|Runtime|207.9|242.7|289.3|330.7|371.8|417.3|454.6|540.2|701.4|
|FLOPs|482.0|516.5|550.9|585.4|619.8|654.3|688.7|757.6|895.4|
|Parameters|1.82|2.87|3.92|4.97|6.02|7.07|8.12|10.22|14.42|
|Memory (test)|4.78|4.92|4.94|4.83|4.97|4.99|5.00|5.03|4.97|
|Memory (train)|56.9|91.7|106.2|122.3|142.3|162.3|185.5|225.5|305.6|

Expectedly, the computational effort increases with a larger number of transformer layers.
A runtime (ms) comparison with other state-of-the-art methods is shown in the following table.
The result shows that the runtime is on par with our strongest competitors: 3DFlow and PT-FlowNet (on the same NVIDIA A40 GPU).
We also report the runtime of other methods for a comprehensive comparison.
|method|GPU|Runtime|method|GPU|Runtime|
|-|-|-|-|-|-|
|FlowNet3D|TITAN V|130.8|FlowStep3D|TITAN RTX|820.8|
|HPLFlowNet|TITAN V|98.4|CamLiFlow|Tesla V100|118.0|
|PointPWC|GTX 1080Ti|117.4|CamLiPWC|Tesla V100|110.0|
|HCRF-Flow|GTX 1080Ti|228.2|CamLiRAFT|Tesla V100|216.0|
|RAFT3D|GTX 1080Ti|386.0|PV-RAFT|NVIDIA A40|740.0|
|FLOT|GTX 2080Ti|389.3|3DFlow|NVIDIA A40|121.6|
|SCTN|GTX 2080Ti|242.7|PT-FlowNet|NVIDIA A40|898.5|
|Bi-PointFlow|TITAN RTX|40.5|**GMSF**|NVIDIA A40|371.8|

**B: Ablation study on the number of transformer layers (D7Yn, YAmp).**

We thank the reviewers for pointing out that the performance may not be saturated with 8 global-cross transformer layers. We initially aimed for a method that works with limited compute resources, in this case 4 NVIDIA A100 GPUs.
We agree that in order to understand our approach better, a larger number of layers should be evaluated. We extend our experiment with more transformer layers. The performance increases with more layers and saturates at a number of 24.
The trade-off are the compute resources as shown in the table above. Detailed results are given in the following table:
|Layers|$EPE_{3D}\downarrow$|$ACC_{S}\uparrow$|$ACC_{R}\uparrow$|$Outliers\downarrow$|$EPE_{3D}\downarrow$|$ACC_{S}\uparrow$|$ACC_{R}\uparrow$|$Outliers\downarrow$|
|-|-|-|-|-|-|-|-|-|
||$all$||||$nonocc$||||
|8|0.049|90.08|94.72|13.08|0.025|94.98|97.78|9.87|
|10|0.046|90.53|94.97|12.57|0.024|95.37|97.92|9.43|
|12|0.046|90.94|95.05|12.20|0.024|95.75|98.03|9.05|
|16|0.041|92.52|95.76|10.31|0.020|96.84|98.38|7.53|
|24|0.041|92.38|95.75|10.16|0.020|96.73|98.39|7.36|

**C: Motivation of Eq. (11) and why it can be viewed as a smoothing procedure (YAmp, vPo9).**

In addition to the explanation in L197-201, we provide further clarification in the following, which will be added to the main paper:

The self-similarity matrix $M_{self}$ bears the similarity information for each pair of points in the source point cloud. Nearby points tend to share similar features and thus have higher similarities. Multiplying $M_{self}$ with the predicted scene flow $\hat{V}_{inter}$ can be seen as an averaging procedure, where for each point, its predicted scene flow vector is updated as the weighted average of the scene flow vectors of the nearby points that share similar features.

**D: Ablation study on occlusion handling / Eq. (13) (YAmp, UZxB, vPo9).**

Given the similarities between optical flow and scene flow, we followed the previous optical flow method RAFT [1] by using Eq. (13) with increasing weights for the refined prediction.

We agree with the reviewers that an additional ablation study on the training loss in Eq. (13) provides further insights and compare three models in the following table as suggested by reviewer vPo9:
Model A, B, and C are trained with only the second term $\hat{V}\_{inter}$, only the first term $\hat{V}\_{final}$, and Eq. (13), respectively.
|Model|$EPE_{3D}\downarrow$|$ACC_{S}\uparrow$|$ACC_{R}\uparrow$|$Outliers\downarrow$|$EPE_{3D}\downarrow$|$ACC_{S}\uparrow$|$ACC_{R}\uparrow$|$Outliers\downarrow$|
|-|-|-|-|-|-|-|-|-|
||$all$||||$nonocc$||||
|A|0.150|73.11|82.91|34.11|0.047|84.37|94.12|24.71|
|B|0.045|91.43|95.38|11.61|0.025|95.36|97.95|8.99|
|C|0.049|90.08|94.72|13.08|0.025|94.98|97.78|9.87|

The result shows that the presence of $\hat{V}_{final}$ in the loss produces good results, which means the occlusion handling helps improve the performance by a large margin.

More interestingly, Model B performs slightly better than Model C. This reveals that scene flow estimation might differ from optical flow estimation regarding the required loss formulation and that supervision of intermediate results is not necessary. (Thanks for the reviewers' comments to help us improve the paper!)

**References**

[1] Teed Z, Deng J. Raft: Recurrent all-pairs field transforms for optical flow[C]//ECCV2020.

---

### Decision · Program_Chairs · 2023-09-21

**Decision:**

Accept (poster)

**Comment:**

This paper receives unanimous recommendation of acceptance. The reviewers acknowledge

- The effective and straightforward model achitecture for scene flow estimation based on Transformer.
- The state-of-the-art performance on standard banchmarks.
- The paper is well written and easy to follow.

Before the rebuttal, some of the reviewers had concerns about FLOPs, GPU memory requirement, runtime, the effectivess of number of Transformer layers, occlusion handling, etc. The authors provided new experimental results, including results on the Waymo dataset, which have addressed the reviewers' comments.

The only limitation of this paper is the limited novelty since the global matching has been adopted for 2D image matching (SuperGlue, LoFTR, and others) and 3D point cloud registration (CoFiNet and GeoTransformer). But the reviewers and AC found it a minor issue.

The AC thus recommends an acceptance as a poster.